# Progress towards a Miniaturised PIV System

**DOI:** 10.3390/s22228774

**Published:** 2022-11-13

**Authors:** Özgün Özer, Mark Kenneth Quinn

**Affiliations:** Department of Mechanical, Aerospace and Civil Engineering, The University of Manchester, Manchester M13 9PL, UK

**Keywords:** PIV, miniaturisation, wind tunnel

## Abstract

Particle image velocimetry is an important optical flow diagnostic tool due to its capacity for investigating a whole flow field without introducing disturbances. However, a significant drawback of PIV methods is their requirement for optical access, making capturing data in closed cavities and confined spaces extremely challenging. A potential approach to overcome this difficulty is miniaturising the system and placing the optical components inside the model. Conventional cross-correlation PIV methods do not allow this due to the size of current PIV cameras. In this study, a miniaturised autocorrelation-based stereo PIV system, which is volumetrically 1.2% of the conventional PIV cameras, was developed and tested. The miniature system is compared with a conventional stereo PIV in wind tunnel experiments up to 16 m/s free stream velocity and a 1.6% velocity difference is observed in the boundary layer flow. Despite a comparatively slow measurement rate of 4.5 Hz, the miniature PIV system demonstrates the ability to measure inside confined spaces and cavities and the ability to be mounted on board models and vehicles. However, limitations remain around conducting measurements with large velocity ranges and with regions of reversed flow due to the challenge of resolving a velocity of 0 m/s.

## 1. Introduction

Since 1984, the particle image velocimetry (PIV) technique has developed into a mainstay of aerodynamic investigation. With its ability to characterise off-surface fluid velocities non-intrusively, PIV provides an attractive solution to many flow measurement scenarios [1]. Besides the creative ideas of several scientists and engineers, the rapid development of the method is also supported by the technological advancement of its subsystems [2]. For example, in the late eighties, the first PIV studies were conducted using film photography. However, the rise of digital imaging vastly enhanced the capability of the technique with regard to spatio-temporal resolution, whilst also improving the practicality of application and data storage [3,4].

Particle illumination has also improved through the development of laser technology, allowing scientists to investigate larger volumes [5,6,7] or measure with higher temporal resolution. In addition to the development of lasers, LEDs are increasingly becoming potential illumination sources in an ever-growing number of PIV scenarios. The illumination output of LEDs is ever increasing whilst simultaneously their costs are decreasing, showing promise for a more affordable, safer, and more compact PIV system [8,9,10,11]. However, the investigation of compact spaces or closed cavities remains a significant challenge. Many engineering applications of PIV do not have good optical access and manufacturing model components from transparent media, enabling optical access, can be impractical due to material strength constraints.

In essence, PIV demands multiple optical access angles to the area of interest in order to illuminate and image the particles. This requirement can be relatively easily achieved when investigating external flows (particularly in laboratory conditions); however, for internal flows, such geometric constraints are hugely limiting and often prevent areas from being investigated. A common solution to the issue of optical access is opening windows to the model or designing a representative model with optical access [12,13,14,15]. In some applications, instead of windows, endoscopic laser optics or cameras are utilised for optical access [16,17,18]. However, in these approaches, the requirement for transparent parts may compromise other aspects of the measurement, such as geometric similarity. For example, when investigating a combustion flow inside a piston, some parts of the piston must be replaced with a transparent material, or there should be holes for endoscopes. Heat transfer to these transparent parts will be significantly different from the rest of the model, potentially introducing an indirect observer effect. Therefore, the question of whether the representative model with optical access reflects the accurate flow physics of the actual system is difficult to answer. Whilst it can be said that endoscopic methods are beneficial for researching a part of the system in detail, they are rarely applicable to a whole working system without extensive modifications.

The miniaturisation of PIV system components is a promising solution for internal investigations, allowing for traditional model design with minimal compromise or modification. These methods aim to utilise non-fluid-containing small internal cavities to install the PIV system without affecting the flow field. The success of this approach is dependent on the miniaturisation of the PIV hardware. As the size of a PIV system is reduced, its range of applicability is greatly expanded. In addition to the research advantages of a miniaturised PIV system, the components used also have the potential to reduce the cost of the system. Combining miniaturised PIV (MPIV) systems with Lab-On-Chip PIV systems might create the opportunity to further reduce cost [19] and conduct in situ experiments.

In 2001 Chételat et al. [20] published a study on designing and constructing a miniature PIV system. In this first study, the authors were able to measure 0.021 m/s velocities with a 288 × 216 resolution camera in the water. A year later [21], the same group presented an upgraded system utilising LED in-line illumination, and increased the measurement velocity up to 1 m/s.

Tritico et al. [22] developed and tested a portable, submersible miniature PIV device in 2007. Images were recorded using a 1 megapixel 10-bit battery-powered CCD camera controlled via a PCMCIA frame grabber card connected to a laptop computer, and the system was validated against a standard laboratory PIV for average velocities up to 0.15 m/s downstream from a 1.6 cm circular cylinder. Another underwater PIV system was presented by Liao et al. in 2009 [23]. The second generation of this system, named a dual-beam dual-camera method for a battery-powered underwater miniature PIV (UWMPIV) system, was developed in 2012 [24] and applied to investigate in situ measurements of sediment resuspension caused by propeller wash [25].

The commercial PIV supplier LaVision has produced a somewhat miniaturised PIV system in a single package known as MiniShaker. This system has been applied to a robotic arm and has been used to map out measurement areas around a model of a cyclist in air by Jux et al. [26]. However, the hardware used in this study is not suitable for use for onboard measurements due to the still relatively large volume required and the relatively long minimum focal distance.

The current work presents the first truly miniaturised PIV system that can measure velocity in the gas medium. The measurement velocity is significantly larger than in previous works, while the system size has been further reduced. The comparison between this system and previous examples from the literature can be seen in Table 1.

In terms of PIV application, air is a significantly more challenging medium than water. The reason for this is the tracer particles used in the application. Since water density is greater than air, larger tracer particles can be used. This allows the use of low-power light sources like continuous lasers or LEDs. However, for investigation in the air, more powerful illumination is needed. Additionally, the velocities experienced in water are also significantly slower than in air. In aerospace research, velocities of hundreds of metres per second are frequently encountered, meaning significantly shorter exposure times are required to image the particles. Such high speeds increase the illumination power requirements as well as more accurate synchronisation of the cameras and illumination system.

The miniaturised system’s ability to work in air shows a substantial improvement upon previous systems. Additionally, air-based measurements have a larger variety of industrial applications, from aerospace to automotive and air-conditioning to household devices.

This positive outcome was achieved by utilising fibre-optic laser cabling with industrial machine vision cameras. The obtained images were post-processed by autocorrelation. Finally, the system was compared and validated against a conventional Stereo PIV (SPIV) system in a low-speed wind tunnel.

## 2. Methodology

In all the experiments presented below, the miniature camera system and laser were present inside the wind tunnel throughout to ensure consistency between the measurements. In this way, any influence of the presence of the in-tunnel hardware would be measured by both systems, enabling fair validation.

### 2.1. Traditional PIV and Validation Experiment Setup

A classical PIV system is typically formed from five subsystems. These can be listed as the seeding system, the illumination system, the imaging system, the synchronisation system, and finally the control and data processing computer.

Most of the fluids investigated are transparent or do not contain optically descriptive patterns due to their homogeneity. Initially, seeding particles with light scattering properties must be added to the fluid as a tracer to make the fluid motion visible. PIV is inherently an indirect measurement method where the researchers do not measure the velocity of the fluid but the velocity of the particles added to the flow. Therefore, it is crucial that the tracer particle size and density must be consistent with the fluid for accurate measurement; otherwise, a velocity lag will occur. As a result, different types of seeding are used for different measurement media and conditions. Solid particles (aluminium flakes or hollow glass spheres) are generally used in liquids or high Mach number gas flows (alumina or titania nanoparticles). Oils such as di-ethyl-hexyl-sebacate (DEHS) or helium-filled soap bubbles are normally used in lower velocity gas measurements.

Micron-diameter particles are normally too small to be visualised directly by the camera; therefore, high-power illumination systems and specific lighting setups are needed to create a contrast with the background to image these particles. Lasers are most commonly used for traditional PIV systems due to their ability to generate ultra-short duration, coherent, high-power illumination. Optics are mounted on these devices to turn the illumination light beam into a plane (referred to as a light sheet or illumination volume), illuminating the tracer particles in the region of interest.

In regular PIV, dedicated PIV cameras with a short interframe duration enable the capture of two sequential images in extremely quick succession (often within 100 s of nanoseconds), which allows for high-speed flows to be imaged clearly. The images from these cameras are calibrated to obtain a scale factor, defining the relation between world and pixel coordinates, and also to remove any distortion from the imaging lens.

This study utilised the University of Manchester’s BOB wind tunnel, which is a low-speed, open-circuit Eiffel-type wind tunnel with a test section measuring 0.9 × 0.9 × 5 m. The boundary layer at 4.5 m along the test section was investigated with both MPIV and regular SPIV systems for validation purposes.

A LaVision SPIV system was used for this validation experiment. The rugged laser (Quantel Evergreen 200 mJ) was mounted to the floor of the wind tunnel with the light sheet aimed vertically, directly upstream along the tunnel centreline. The cameras were positioned outside of the wind tunnel and image the flow through large transparent windows, as shown in Figure 1.

A 106 mm × 106 mm double-layer calibration plate, which has 2.2 mm dots with 10 mm spacing, was used to calibrate the cameras. Two hundred image pairs were captured for each camera in every experiment. After the experiment, a Planar Self Calibration process was applied to the images to remove any residual error from the positioning of the calibration plate and the actual position of the light sheet. DaVis software 10.2.1.80999 was used for the calibration process, image pre-processing, and post-processing of the SPIV images.

The minimum intensity value of each pixel was calculated through all of the data sets and subtracted from all images for background image removal. A local minimum subtraction filter with 3 × 3 px was applied for contrast enhancement. Subsequently, multi-pass cross-correlation with two stages was applied with a 64 × 64 px initial interrogation area, reducing to 32 × 32 px for the final pass. A 75% window overlap was used during the correlation of the instantaneous calculations, resulting in 200 vector maps for each condition. Finally, the average of these vectors was calculated to obtain the average flow structure.

The specifications of the components for SPIV are listed in Table 2.

### 2.2. Miniaturised PIV (MPIV) System

In terms of miniaturisation, camera systems are the most critical as they require the most extensive optical access. As mentioned, camera endoscopes can supply optical access, but since they are relatively short, the cameras are generally positioned externally to the model. Therefore, the external flow of the system cannot be tested simultaneously. Additionally, the cases with a flow interaction between the internal and external flow cannot be tested. MPIV offers a solution to this situation.

Two Ximea MU9PM-MH cameras, which measure 15 × 15 × 8 mm^3^, were used for the MPIV with 6 mm f8 lenses mounted on them. These cameras were installed on a carrier beam and positioned inside the wind tunnel, as shown in Figure 2. The same calibration process with planar self-calibration was also applied in the MPIV experiments. The regular PIV cameras were 61 × 80 × 80 mm^3^ in volume without a lens attached, clearly showing the reduction in volume of the miniature system; however, the achievable capture frame rate of the camera was decreased (4.5 Hz for the miniature camera compared to 31 Hz for the regular PIV camera).

However, there is a significant difference between the dedicated PIV and machine vision cameras used in MPIV. As mentioned previously, dedicated PIV cameras have the ability to interframe, enabling imaging of each laser pulse within a separate frame. An example of such imaging is given in Figure 3.

The machine vision cameras utilised in this study do not have interframe capability. Therefore, two pulses were captured in a single frame, as shown in Figure 4. However, these miniature cameras do have global reset release capability, enabling the exposure of all pixels to happen simultaneously.

As a result of the lack of interframe capability, autocorrelation must be used instead of the standard cross-correlation processing methods of traditional PIV. In this study, a multi-pass autocorrelation algorithm was applied with initial interrogation areas of 64 × 64 px, a final window size of 32 × 32 px, and a window overlap of 50%. A major drawback when using autocorrelation is the directional ambiguity of the result. Since both pulses are captured in the same frame, there is no way to determine the order of the pulses and therefore vector direction without additional information. If a priori information is known, the autocorrelation algorithm can be steered to select the correct displacement peak. For example, the flow in Figure 5 is known to be from left to right, meaning the autocorrelation output can be corrected.

In this method, similar to the definition of boundary conditions in Computational Fluid Dynamics (CFD), the directions of the known locations are defined. The direction of other positions can be evaluated according to these known locations by a moving filter or other methods. It must be noted that this moving filter does not change the magnitude of the vector; it only influences the direction. The method can be applied iteratively until a coherent vector map is obtained. However, as will be demonstrated, this method struggles to resolve complex vortex structures and reversed flow.

The other essential component that should be miniaturised is the illumination system, as it also requires optical access. In miniaturising studies in the literature, low-power continuous lasers or LEDs are used for this purpose. However, this approach limits the illumination power, fluid medium, and flow velocity. In this study, the light of a standard PIV laser is carried by a fibre-optic laser guide. The fibre guide is a round-to-linear bundle array of individual fibres which have a high-aspect-ratio cylindrical lens as a sheet-forming optic at the exit. Although the fibre-optic cable only permits 50 mJ per pulse, this per-pulse energy is still significantly greater than the previously used alternatives.

As autocorrelation is being used for this investigation, the laser pulses can be triggered independently and still be captured in one image, meaning there is essentially no upper limit to the velocity that can be measured. The improvements in optical power and timing enabled the investigation of a gas medium and fast flows with this novel MPIV system. The cameras and the fibre-optic laser guide can be mounted on a carrier platform for more compact packaging. An example of this is given in Figure 6.

**Figure 6 sensors-22-08774-f006:**
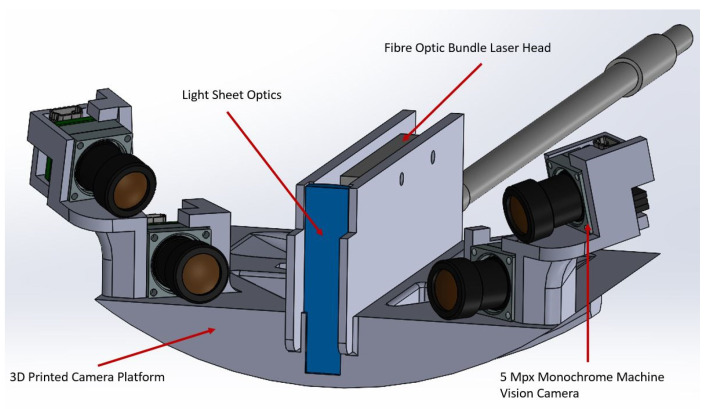
Compact packaging of the cameras and the fibre-optic laser guide fitted inside a working wind tunnel model of a jet engine [27].

The same seeding generator that bulk-seeds the wind tunnel is also utilised for both miniature and regular PIV experiments. The components of the MPIV and their specifications are listed in Table 3.

## 3. Results and Discussion

Although MPIV systems are being developed for closed cavity flow, the validation experiment was conducted on external flows. This is because traditional PIV systems cannot be used in small, closed cavities. Therefore, the boundary layer at 4.5 m of the test section of a low-speed wind tunnel was investigated at five freestream velocities with both systems for validation. The time between pulses ($\delta_{t}$) of the laser was adjusted for different freestream velocities to keep the pixel displacement the same for each experiment. A sample raw image captured by MPIV is shared in Figure 7. The nominal velocity of the wind tunnel (U_∞_), the Reynolds number and the $\delta_{t}$ values of the test conditions are given in Table 4.

The vector maps obtained by SPIV and MPIV are given in Figure 8. Due to the optical arrangement (sensor size and lens focal length), the SPIV captured a larger area of the flow. The velocity distribution captured with both systems seems largely coherent; however, there are differences in near-wall areas, as shown in Figure 9.

A likely explanation for the deviation between the techniques is the limited dynamic range of autocorrelation and its inability to measure zero displacement. In this method, since the two pulses are captured in the same image, as the velocity and the pixel displacement decrease, the particles start to get closer and, ultimately, merge. When overlap exists between the positions of the particles in both laser pulses, it is almost impossible to distinguish the particles, and the correlation map peak is too close to the 0.0 value. This overlap removes any opportunity to recover a velocity measurement.

The average difference between the flow profiles of the two measurement methods is 1.6%.

An additional validation test was conducted further to investigate the performance of the MPIV system on more challenging flows. A circular cylinder was placed horizontally in the same test area investigated previously. The cylinder had a Reynolds number of 22,000 and was investigated with both MPIV and SPIV methods. The raw image and close-ups are given in Figure 10.

Comparing close-up images, it can be seen that the wake of the cylinder is not well resolved. The physical size of the interrogation window means that the strong gradients in the wake region cannot be well resolved. As both particle positions are captured in the same image, autocorrelation finds this scenario significantly more challenging than a boundary layer flow. However, this limitation is not present when using cross-correlation, as demonstrated by the vector maps obtained from both methods (see Figure 11).

MPIV did not perform as well in this case due to autocorrelation constraints when investigating reversed flow. Since the Von Karman vortex street generated velocities ranging between −6 and 22 m/s, the dynamic range limitation of the autocorrelation failed to capture velocity characteristics in the wake of the cylinder. As mentioned previously, due to particle overlap when displacements are small, measuring velocities close to zero is not possible with this approach. Figure 12 shows the instantaneous vector fields obtained from both methods as a further demonstration of this phenomenon. Although the magnitude of the separated flow was captured relatively well, further improvement is needed for the investigation of highly turbulent or reversed flows.

## 4. Conclusions

A miniaturised PIV system has been developed that can be situated within a wind tunnel model. This system utilises a combination of off-the-shelf components and custom-made hardware to facilitate miniaturisation. The system has been validated against a commercially available PIV system and has the ability to capture high-speed air flows.

The system gives accurate results for fully attached flows or other simple cases where directional ambiguity can be solved effectively. A turbulent boundary layer measurement was achieved with a 1.6% difference from a conventional PIV system. However, in highly recirculating flows, although the velocity magnitude is obtained relatively well in some areas, directional ambiguity and the inability of autocorrelation to resolve small displacements proved to be a significant problem. Therefore, alternative methods such as multi-pulse scheduling or more advanced methods for extracting and guiding correlation should be applied in future studies.

The ability to measure velocities close to zero is hampered due to the overlap of the particles in both exposures. An iterative approach to schedule different pulse separations for different parts of the flow field could help resolve this challenge and is being investigated in future studies by the authors.

In this study, the system was tested up to 16 m/s; however, given the available time between laser pulses and the camera resolution, theoretically much higher velocities can be resolved. This will be tested in future studies.

The miniaturisation of the synchronisation system and the control computer might have advantages for in situ testing, and a complete System on Module (SOM) (such as Nvidia Jetson Nano) is capable of controlling the MPIV system. In future studies, this type of control might offer further size reduction or enhance flexibility for onboard measurements. This system opens the door to industrial use of MPIV and provides an important tool for researchers to investigate closed cavity flows or onboard measurements of real vehicles.

The work presented here was completed within the TRUflow project as a part of the European Commission Clean Sky 2 research and innovation scheme. As the study’s next step, this system will be implemented in a working wind tunnel model of a jet engine. If the method succeeds, it will be utilised to visualise and evaluate reverse flow interactions with fan aerodynamics in short, slim engine nacelle designs. These designs are directly applicable to the next generation of ultra-high bypass ratio (UHBR) aero engines which aim to reduce CO_2_ emissions.

The development of such industrial sensors supports researchers and engineers in achieving the UN’s sustainable development goals, such as industry innovation and infrastructure (9), sustainable cities, transportation and communities (11), and climate action (13).

## Figures and Tables

**Figure 1 sensors-22-08774-f001:**
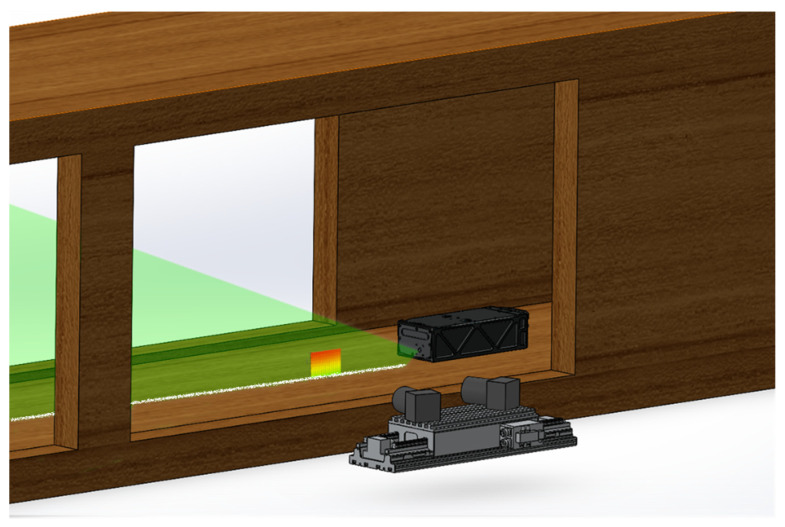
Camera and laser positions of the SPIV system in wind tunnel tests.

**Figure 2 sensors-22-08774-f002:**
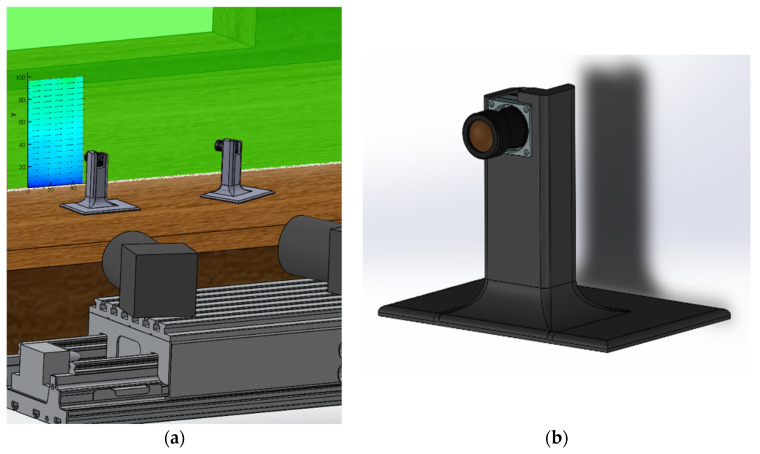
The internal PIV setup: (**a**) The camera positions in the wind tunnel; (**b**) close-up image of the camera and the carrier beam; (**c**) size comparison of the cameras.

**Figure 3 sensors-22-08774-f003:**
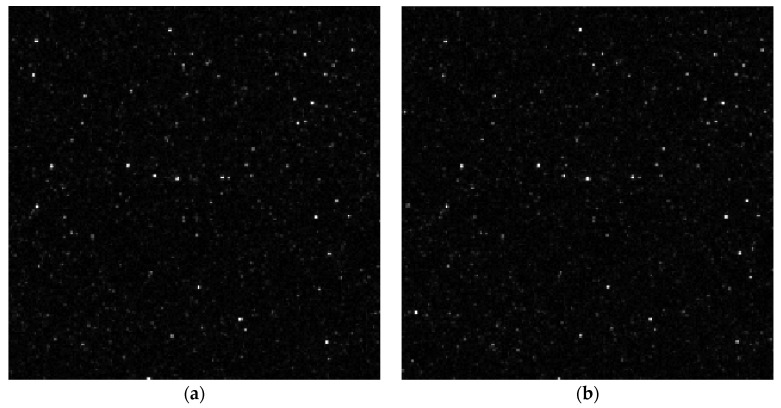
The sample close-up particle images were captured by the traditional PIV camera: (**a**) The image of the first pulse; (**b**) the image of the second pulse.

**Figure 4 sensors-22-08774-f004:**
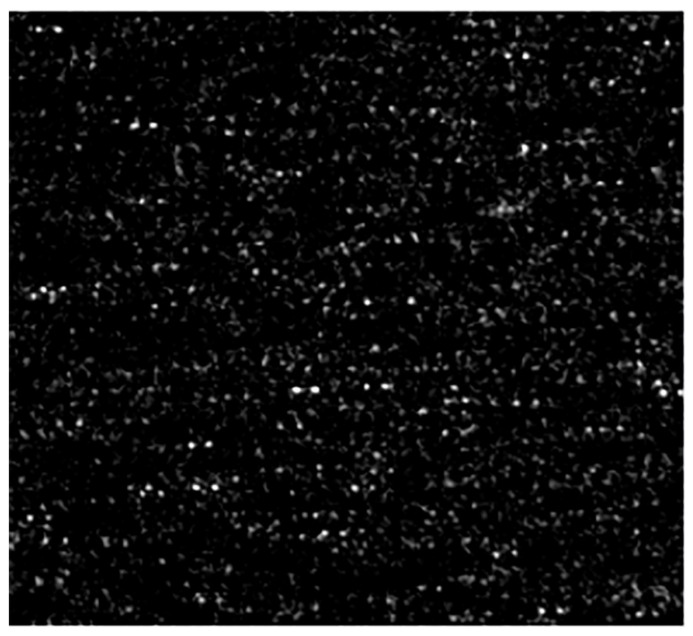
The sample close-up particle image was captured by the machine vision camera.

**Figure 5 sensors-22-08774-f005:**
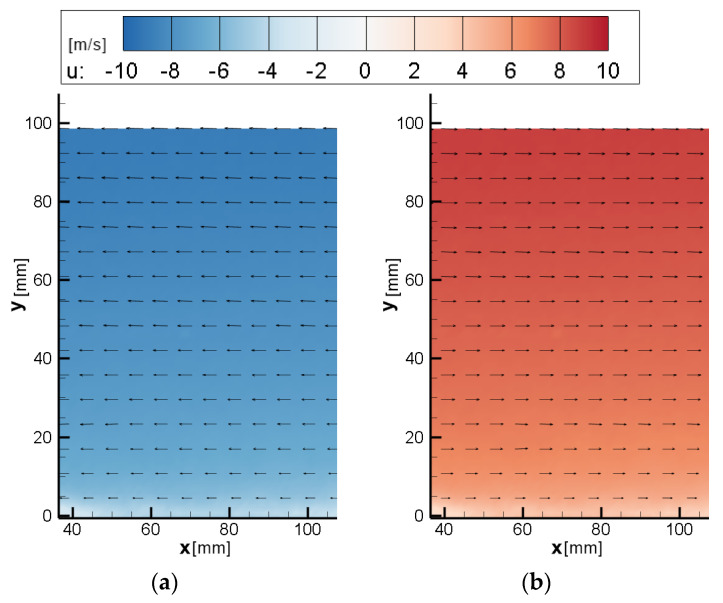
Directional ambiguity example of the wind tunnel boundary layer at 8 m/s: (**a**) result with directional ambiguity; (**b**) corrected by post-processing.

**Figure 7 sensors-22-08774-f007:**
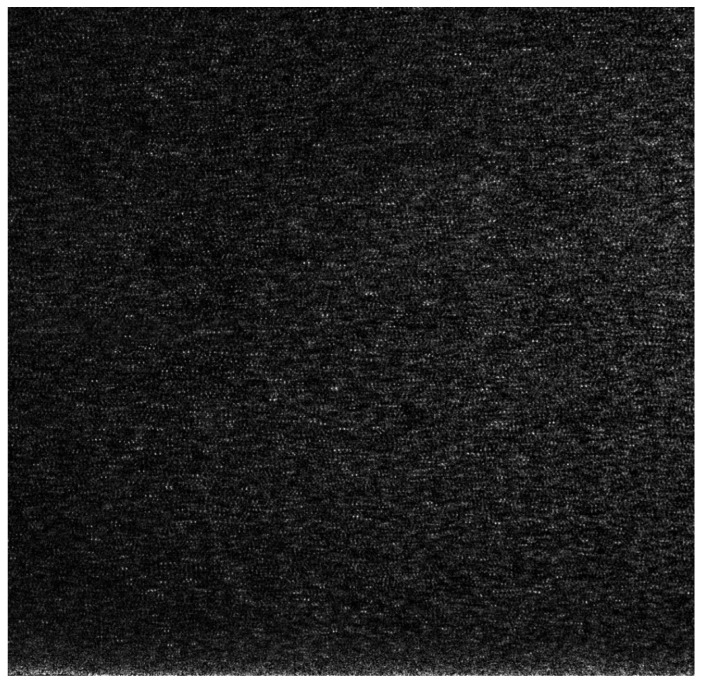
Raw MPIV image captured during C1 test.

**Figure 8 sensors-22-08774-f008:**
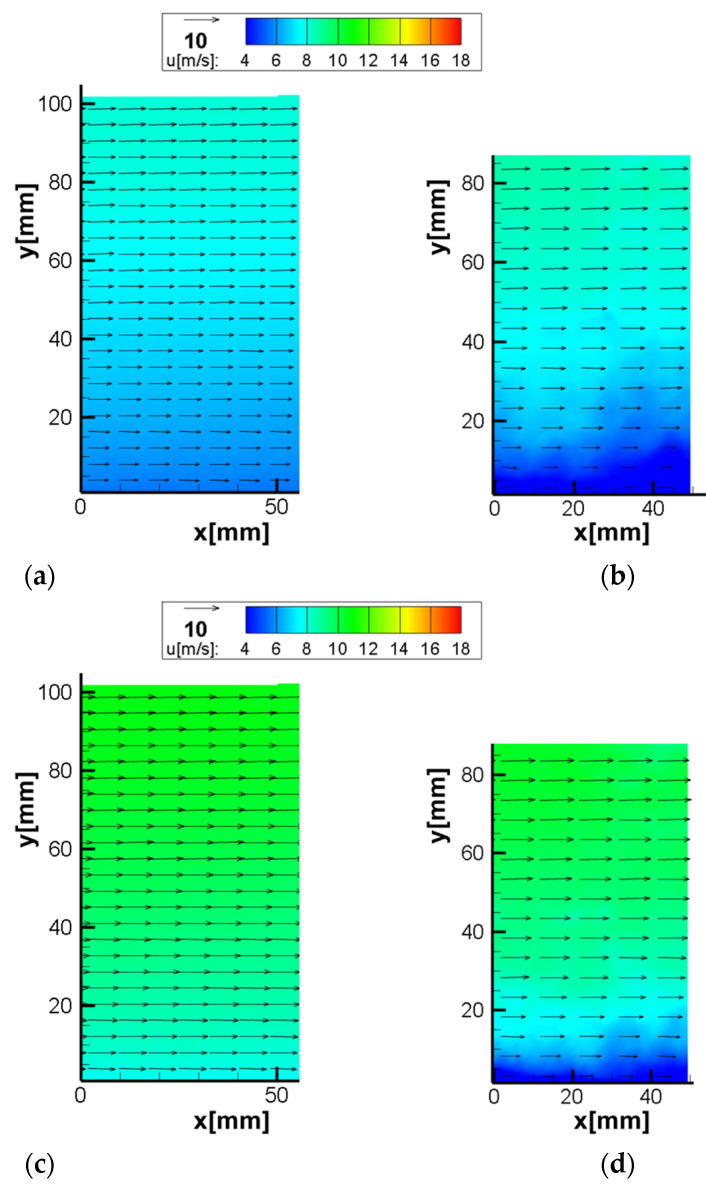
Vector map results of the boundary layer test (**a**) C1 SPIV $U_{\infty }=8$ m/s; (**b**) C1 MPIV $U_{\infty }=8$ m/s; (**c**) C2 SPIV $U_{\infty }=10$ m/s; (**d**) C2 MPIV $U_{\infty }=10$ m/s; (**e**) C3 SPIV $U_{\infty }=12$ m/s; (**f**) C3 MPIV $U_{\infty }=12$ m/s; (**g**) C4 SPIV $U_{\infty }=14$ m/s; (**h**) C4 MPIV $U_{\infty }=14$ m/s; (**i**) C5 SPIV $U_{\infty }=16$ m/s; (**j**) C5 MPIV $U_{\infty }=16$ m/s.

**Figure 9 sensors-22-08774-f009:**
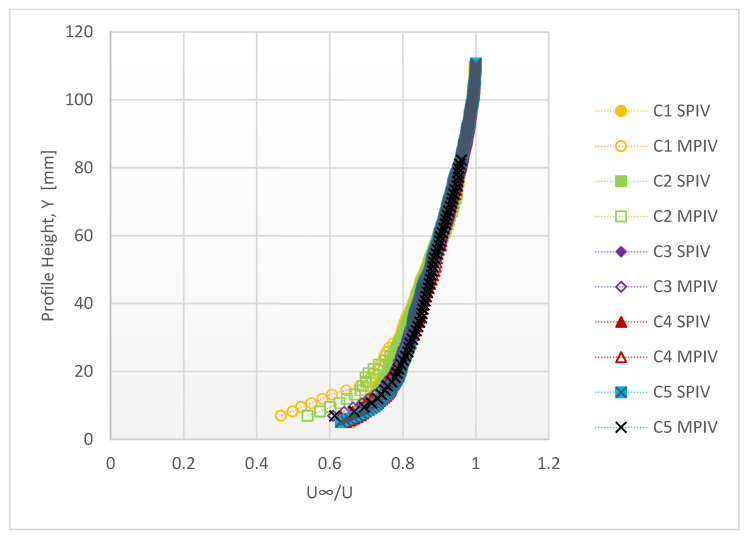
Velocity profiles measured in the boundary layer experiment.

**Figure 10 sensors-22-08774-f010:**
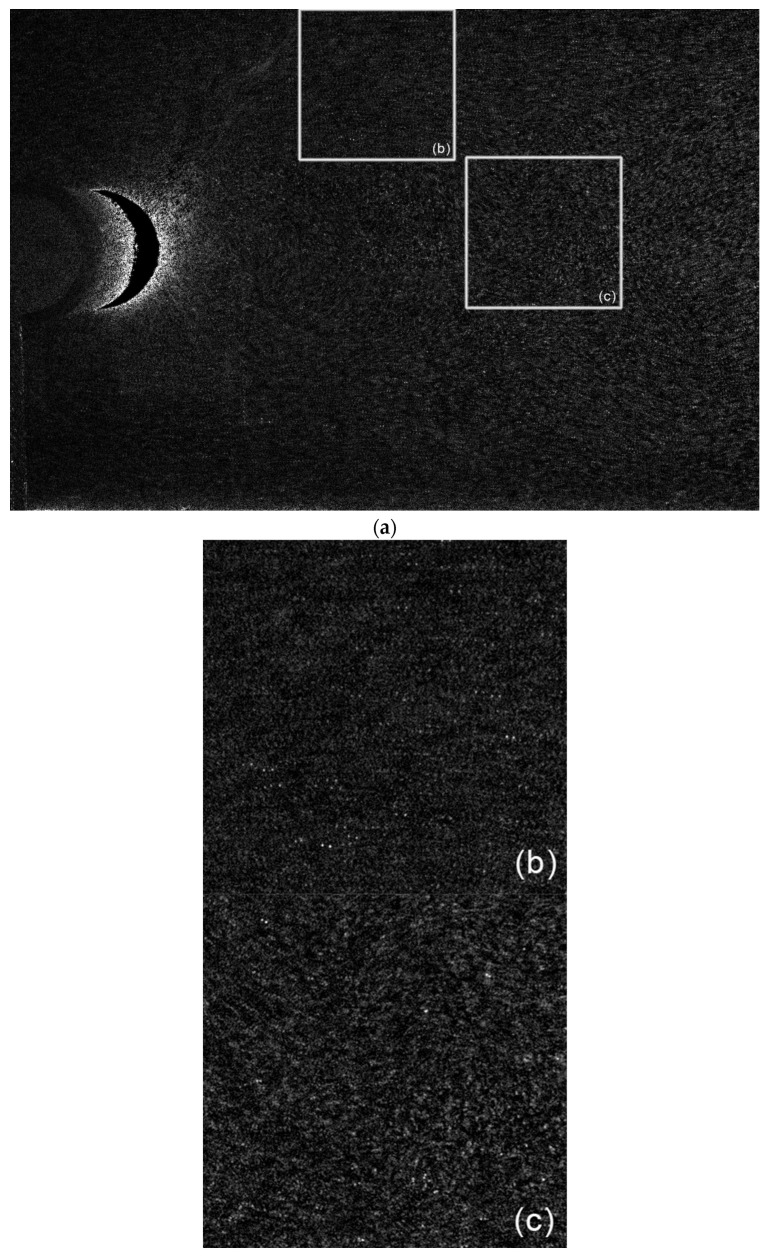
Raw images of MPIV during the second experiment: (**a**) full image; (**b**) close-up at the separation area; (**c**) close-up in the wake of the cylinder.

**Figure 11 sensors-22-08774-f011:**
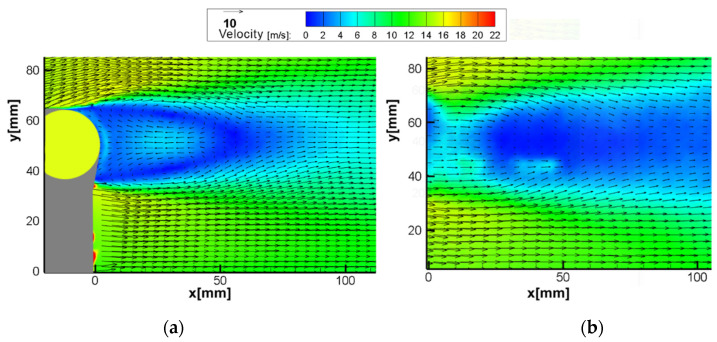
Vector map results of the flow around a cylinder test (Re = 22,000): (**a**) SPIV; (**b**) MPIV.

**Figure 12 sensors-22-08774-f012:**
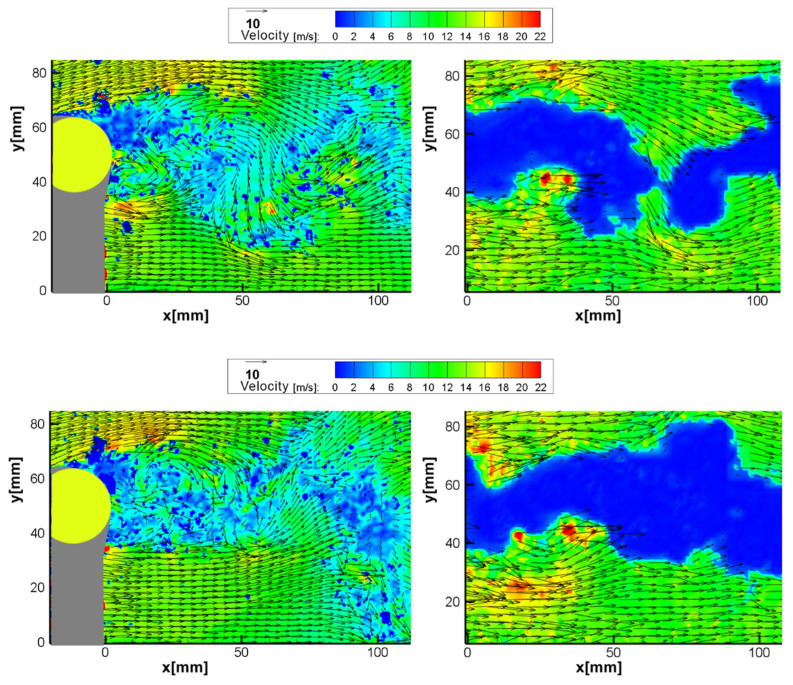
Instantaneous vector map results of the flow around a cylinder test (Re = 22,000): (**a**) SPIV; (**b**) MPIV.

**Table 1 sensors-22-08774-t001:** Comparison of the MPIV systems.

MPIV System	Measurement Medium	Measured Velocity [m/s] *	Camera Resolution	Camera Size [mm]
Chételat et al. (2001) [20]	Water	0.021	288 × 216	140 **
Chételat et al. (2002) [21]	Water	1	288 × 216	140 **
Tritico et al. [22]	Water	0.15	1024 × 1024	50 × 39 × 83
Wang et al. [24]	Water	0.50	1360 × 1024	59 × 46 × 33
Liao et al. [25]	Water	0.6	1360 × 1024	59 × 46 × 33
Jux et al. [26]	Air	16 ^†^	1920 × 1280	135 × 110 × 319 ^‡^
Present Study	Air	16	2592 × 1944	15 × 15 × 8

* Maximum Measured Velocity of the study is selected. ** The size of the system was not mentioned in the articles. However, in an image on the paper, a ruler showed that one dimension of the system is around 140 mm. ^†^ Maximum measurement velocity is a function of field of view and the minimum interframe time which is ≈60 µs. ^‡^ Measurement unit size not individual camera.

**Table 2 sensors-22-08774-t002:** Components and specifications of the SPIV system.

Component	Specifications
Laser	Evergreen 532 nm, ND:YAG Laser 200 mJ Maximum power per pulse
Cameras	2 × LaVision SX4 Mpx. (2360 × 1776 px)
Lens	Tokina, AT-X M100 PRO D Macro 100 mm (15.54 px/mm Scale factor)
Software	LaVision DaVis 10.2.1.80999
Calibration Plate	106-10 Double Layer Calibration Plate
Seeding	DEHS delivered by the atomiser

**Table 3 sensors-22-08774-t003:** Components and specifications of MPIV system.

Component	Specifications
Laser	Evergreen 532 nm, ND:YAG Laser (200 mJ Maximum power per pulse) Delivered by custom-made fibre optic laser guide (50 mJ Maximum power per pulse)
Cameras	Ximea MU9PM-MH, 5 Mpx
Lens	6 mm f8 M12 Lens (12.75 px/mm scale factor)
Software	For control and synchronisation: LaVision DaVis 10.2.1.80999 For post processing: MATLAB
Calibration Plate	106-10 double-layer calibration plate
Seeding	DEHS delivered by the atomiser TSI9306A

**Table 4 sensors-22-08774-t004:** Conditions of the boundary layer test.

Condition Index Number	Nominal Velocity $U_{\infty }$ [m/s]	Reynolds Number $\left(\times 10^{6}\right)$	$\delta_{t}$ [µs]
C1	8	2.38	80
C2	10	2.97	64
C3	12	3.56	53
C4	14	4.16	46
C5	16	4.75	40

## Data Availability

Not applicable.

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
