# Peer review of "Progress towards a Miniaturised PIV System"

_sensors, 2022, doi:10.3390/s22228774_

Round 1

Reviewer 1 Report

The paper discussed a miniaturised PIV system, which is of great intrested toward a benefit to fuild experiments. Some questions and suggestions with regard to this paper is listed below.

(1) Line 93 was ended with a comma. The paragraph seems incompleted. Please improve.

(2) It seems that the major improvent on the PIV system is the size of camera. Can you mark the size of the cameras (conventional and improved) on Fig. 2? 

(3) Furthermore, if minimizing the size of camera brings any performance deficiency, this should be mentioned. Please add a table and discussion that compares the proposed camera with conventional ones, including some key performance parameters.

(4) Lines 274-280, is this paragraph discussing Fig. 12? The disscussion on Fig. 12 is missing. This should be the limitaion of the proposed system, more quantified information are required.

(5) The application scopes, limitaions, ect. of the proposed system should be mentioned in abstract and conclusion chapters.

Author Response

(1) Line 93 was ended with a comma. The paragraph seems incompleted. Please improve.

R1- Table 1 was duplicated by mistake during page formatting. The duplicate version is removed from the manuscript.

(2) It seems that the major improvent on the PIV system is the size of camera. Can you mark the size of the cameras (conventional and improved) on Fig. 2?

A sentence has been added above Figure 2 describing the comparative sizes of the two imaging systems. Additionally, a photograph of two cameras was added in Figure 2 c to show this comparison.

(3) Furthermore, if minimizing the size of camera brings any performance deficiency, this should be mentioned. Please add a table and discussion that compares the proposed camera with conventional ones, including some key performance parameters.

The only pertinent difference between the two cameras other than size and the interframing ability is the frame rate – this has been added on line 194

(4) Lines 274-280, is this paragraph discussing Fig. 12? The disscussion on Fig. 12 is missing. This should be the limitaion of the proposed system, more quantified information are required.

Line 277, Figure 11 should be figure 12. This is now line 313 and has been corrected. A sentence has been added linking back to the previous discussion on the challenges of measuring small displacements with autocorrelation. However, it is difficult to quantify the limitations in this way as the resulting minimum displacements will depend on the optical arrangement, the scale and size of the particles and also the pulse separation. Suggestions for alleviating these issues are made in the conclusion and are currently under investigation.

(5) The application scopes, limitaions, ect. of the proposed system should be mentioned in abstract and conclusion chapters.

A sentence has been added to the abstract (line 17) to describe the measurement rate (one of the limitations) and the limitations of measuring 0 m/s. Also, a description of applications such as onboard measurements and closed cavities has been added. Sentences describing the challenges of resolving 0 m/s (along with potential solutions) have been added to the conclusion (line 343)

Reviewer 2 Report

This manuscript reports a newly designed miniaturized PIV. Some potential drawbacks from such a system are well addressed; the use of multipass autocorrelation technique and fiber optic laser is applied in this design to overcome such drawbacks. A validation in the wind tunnel by contrasting it to the commonly used stereo PIV for boundary-layer flow and flow over a cylinder. The PIV specifications and validations are presented in a comparatively straightforward and concise manner.

My specific comments are:

1. Please add a sentence or two highlighting the validation results of this work in the Abstract. The authors can discuss the turbulent boundary layer measurement; maximum free stream velocity of 16 m/s, and/or the 1.6% difference from the SPIV.

2. Line 67: I guess the reference for “A year later …” should be [20], not [15].

3. Lines 90-91 should be removed.

4. Figure 1: It would have been preferable to include the full sketch rather than just this up-close photo. Please also specify the region of interest and the origin of the coordinates used (such as in Figs. 5 and 8) in the sketch. Is x = 0 mm at 50 cm upstream of the end of the test section? How can it be made sure that the SPIV laser's positioning and the test section's open end have no impact on the flow?

5. Where is the exact spanwise location of the MPIV cameras? As they were positioned inside the test section, how could the authors ensure they wouldn’t obstruct the main flow?

6. Line 277: Figure 12, not 11.

7. Figure 6: it seems the supposedly “3D printed camera platform” is cropped awkwardly.

Author Response

  1. Please add a sentence or two highlighting the validation results of this work in the Abstract. The authors can discuss the turbulent boundary layer measurement; maximum free stream velocity of 16 m/s, and/or the 1.6% difference from the SPIV.

Suggested update applied to the abstract. (Line 16)

  1. Line 67: I guess the reference for “A year later …” should be [20], not [15].

 This has been corrected in the text.

  1. Lines 90-91 should be removed.

 Table 1 was duplicated by mistake during page formatting. The duplicate version is removed from the manuscript.

  1. Figure 1: It would have been preferable to include the full sketch rather than just this up-close photo. Please also specify the region of interest and the origin of the coordinates used (such as in Figs. 5 and 8) in the sketch. Is x = 0 mm at 50 cm upstream of the end of the test section? How can it be made sure that the SPIV laser's positioning and the test section's open end have no impact on the flow?

The total length of the wind Tunnel is around 15.6m, and the cameras are 15mm in size. Because of this scale difference, it is visually impossible to show the setup in the full-scale wind tunnel model. For this reason, the test section of the wind tunnel is shown in the image. As indicated in the manuscript, the wind tunnel is Eiffel Type which has the fan at the end of the tunnel. The location of the test section is around 6.5m to the end of the wind tunnel.

It is not expected that the positioning of the laser will have much impact on the flow in the tunnel as it is 500 mm downstream of the measurement area. Regardless of the position of the laser, both techniques are measuring the exact same spatial locations with any upstream influence of the laser being present for both.

Figure 1 and 2a have been updated to improve the clarity of the measurement location and the position of the measurements was added to the figures. A larger section of the wind tunnel is also presented in figure 1 showing that the tunnel does not abruptly end as shown previously.

  1. Where is the exact spanwise location of the MPIV cameras? As they were positioned inside the test section, how could the authors ensure they wouldn’t obstruct the main flow?

The cross-section of the wind tunnel is 920 mm x 920 mm. The camera holder is 80mm high with a 20 mm x 20 mm cross section. The blockage caused by the cameras is approximately 0.002 % of the cross-section. The first camera is placed in the middle of the investigation area in the flow direction, and the second camera is placed downstream of the examined area. The cameras are placed 220 mm from the measurement plane (in the spanwise direction), and since the camera holder’s horizontal length is 20mm, vortex shedding of the camera holder cannot be large enough to affect the investigation plane.

The miniature camera were mounted inside the wind tunnel when the regular PIV system measurement was taking place so any influence will be present for both systems, therefore not having an impact on validation. A short paragraph has been added to explain this at the beginning of the methodology.

  1. Line 277: Figure 12, not 11.

This error has been corrected.

  1. Figure 6: it seems the supposedly “3D printed camera platform” is cropped awkwardly.

The figure has been improved.

Reviewer 3 Report

The authors developed and tested a miniaturised autocorrelation-based stereo PIV system which is volumetrically 1.2 % of the conventional PIV cameras, and the miniature system is compared with a conventional stereo PIV in wind tunnel experiments. The work is well done, and the discussion is sufficient. I think it can be published after minor revision.

1.      In the introduction part, the references are relatively old, and resent researches that provide new development in the field of PIV should be detailed and discussed to make the background more sufficient.

2.      No explanation was done for Fig. 12.

3.      Please provide more clear figure for Figure 4 and Figure 7.

Author Response

  1. In the introduction part, the references are relatively old, and resent researches that provide new development in the field of PIV should be detailed and discussed to make the background more sufficient.

Unfortunately, the Miniaturized PIV literature in this paper is up to date, there has been little progress in the area over the past 10~15 years.  However, the literature expanded with new studies on PIV and other aspects of PIV and reference has been made to a commercial system which combines cameras and illumination into one head.

  1. No explanation was done for Fig. 12.

Figure 11 should be figure 12. The error has been corrected.

  1. Please provide more clear figure for Figure 4 and Figure 7.

Figures replaced with higher resolution versions.

Reviewer 4 Report

In this article, a miniaturised autocorrelation-based stereo PIV system was developed and tested. The obtained results are interesting, but the paper cannot be published in the current version. There are several suggestions:

1. Table 12 seems to be the same as Table 1, please explain.

2. Some words are missing in Figure 6, which makes the reviewer confusing.

3. In Table1, previous works all focus on water, but the author’s method can measure velocity in the gas medium, why these methods can be compared with a different medium? Please explain.

4. The references are relatively out-of-date, some more new papers within three years should be cited.

5. There are many grammatical errors, and the author should tackle the problems of the English tense.

Author Response

  1. Table 12 seems to be the same as Table 1, please explain.

R1-Table 1 was duplicated by mistake during page formatting. The duplicate version is removed from the manuscript.

  1. Some words are missing in Figure 6, which makes the reviewer confusing.

R2- The figure has been corrected.

  1. In Table1, previous works all focus on water, but the author’s method can measure velocity in the gas medium, why these methods can be compared with a different medium? Please explain.

The following explanation was also added to the text. (Line 102)

In terms of PIV application, air is a significantly more challenging medium than water. The reason for this is the tracer particles used in the application. Since water density is greater than air, larger tracer particles can be used. This allows the use of low-power light sources like continuous lasers or LEDs. However, for investigation in the air, more powerful illumination is needed. Additionally, the velocities experienced in water are also significantly slower than in air. In aerospace research, velocities of hundreds of metres per second are frequently encountered, meaning significantly shorter exposure times are required to image the particles. Such high speeds increase the illumination power requirements as well as more accurate synchronisation of the cameras and illumination system.

The miniaturised system’s ability to work in air shows a substantial improvement upon previous systems. Additionally, air-based measurements have a larger variety of industrial applications, from aerospace to automotive and air-conditioning to household devices.

Some studies have utilised larger particles in air such as helium filled soap bubbles (a paper by Jux et al. is referenced) enabling lower illumination levels; however, the issue of miniaturisation of imaging hardware remains. Additionally, these bubbles pop above certain flow speeds/ pressure changes.

Our system’s ability to work in the air shows a substantial improvement from the previous systems particularly in terms of the flexibility of measurement it can enable. Additionally, the air medium has a larger variety of industrial applications, from aerospace to automotive, air-conditioning to household devices.

  1. The references are relatively out-of-date, some more new papers within three years should be cited.

Unfortunately, the Miniaturized PIV literature in this paper is up to date, there has been little progress in the area over the past 10~15 years.  However, the literature expanded with new studies on PIV and other aspects of PIV and reference has been made to a commercial system which combines cameras and illumination into one head.

  1. There are many grammatical errors, and the author should tackle the problems of the English tense.

A thorough review and extensive edit has been made on that paper and it now reads much better than the previous version.

Round 2

Reviewer 1 Report

The manuscript is ready for publication.

Reviewer 4 Report

No more comments